# Experimental Investigation on Mean Flow Development of a Three-Dimensional Wall Jet Confined by a Vertical Baffle

**Ming Chen [1,2,*], Haijin Huang [1], Xingxing Zhang [1], Senpeng Lv [1] and Rengmin Li [1]**

[1]  Key Laboratory of Hydraulic and Waterway Engineering of the Ministry of Education, Chongqing Jiaotong University, Chongqing 400074, China; hhj910219@163.com (H.H.); zhangddy@126.com (X.Z.); sinplor@126.com (S.L.); lrengmin@126.com (R.L.)

[2]  Key Laboratory of Navigation Structure Construction Technology, Ministry of Transport, Nanjing 210029, China

*  Correspondence: chenmingjy@126.com; Tel.: +86-023-62652714

**Abstract:** Three-dimensional (3D) confined wall jets have various engineering applications related to efficient energy dissipation. This paper presents experimental measurements of mean flow development for a 3D rectangular wall jet confined by a vertical baffle with a fixed distance (400 mm) from its surface to the nozzle. Experiments were performed at three different Reynolds numbers of 8333, 10,000 and 11,666 based on jet exit velocity and square root of jet exit area (named as *B*), with water depth of 100 mm. Detailed measurements of current jet were taken using a particle image velocimetry technique. The results indicate that the confined jet seems to behave like an undisturbed jet until 16*B* downstream. Beyond this position, however, the mean flow development starts to be gradually affected by the baffle confinement. The baffle increases the decay and spreading of the mean flow from 16*B* to 23*B*. The decay rate of 1.11 as well as vertical and lateral growth rates of 0.04 and 0.19, respectively, were obtained for the present study, and also fell well within the range of values which correspond to the results in the radial decay region for the unconfined case. In addition, the measurements of the velocity profiles, spreading rates and velocity decay were also found to be independent of Reynolds number. Therefore, the flow field in this region appears to have fully developed at least 4*B* earlier than the unconfined case. Further downstream (after 23*B*), the confinement becomes more pronounced. The vertical spreading of current jet shows a distinct increase, while the lateral growth was found to be decreased significantly. It can be also observed that the maximum mean velocity decreases sharply close to the baffle.

**Keywords:** experiment; particle image velocimetry; 3D confined wall jet; mean flow

## 1. Introduction

It is well understood that three-dimensional (3D) wall jets are typically characterized by the interaction between a turbulent boundary layer and a free jet. Because of their diverse practical engineering applications (e.g., film cooling [1], heat transfer [2], and energy dissipation in hydraulic structure [3–8]), a number of experimental studies on the mean flow development in the jet have been conducted in the past few decades. The experiments of 3D wall jet issuing from rectangular orifices have been firstly performed by Sforza and Herbst [9]. From their results, it is well known that the flow field of jet can be divided into three regions: the potential core region (PC), the characteristic decay region (CD) and the radial decay region (RD). Subsequent to their reviews, considerable investigations were carried out for the unconfined case. For example, Padmanabham and Gowda [10] measured the mean flow characteristics of 3D wall jets using a technique of the total pressure probe and determined

the influence of the geometry on the characteristic decay region. Law and Herlina [11] investigated the velocity and concentration characteristics of 3D turbulent circular wall jets using a combined PIV and Planar Laser Induced Fluorescence approach. The results showed that velocity profiles collapsed well in both the longitudinal and lateral directions after 20 nozzle diameters and 25 nozzle diameters, respectively. Additionally, the spreading of the jet and decay of local maximum velocity were critically presented. Agelin-Chaab and Tachie [12] performed PIV (particle image velocimetry) measurements for the 3D wall jet issuing from a square nozzle. The main goal of their measurements was to examine the effect of the rough surface on the flow development of the jet. Later, they presented more detailed laboratory investigations on the jet, including the mean flow in the developing and self-similar regions [13]. Recently, detailed flow structures of a 3D curved wall jet have been reported by Kim et al. [14]. The results indicated that, due to the Coanda effect [15], the jet developed on the cylinder surface after the impingement of the circular jet and self-preserving wall jet profile did not clearly occur in such jet.

In the past, there have been only a few studies on the 3D confined wall jet. In order to determine whether the large lateral growth of the jet is induced by the secondary flow, Després and Hall [16] measured the flow field in a 3D wall jet with and without the grid using hot-wire anemometry and PIV. They found that the grid delayed the lateral growth of the jet and increased its vertical growth. Meanwhile, the grid also decreased the mass entrainment and mixing performance. The grid, however, was placed at the nozzle exit in their studies; thus, the confinement condition differs significantly from that in the present study. Onyshko et al. [17] provided the PIV data for a deflected wall jet. The results showed that the baffle placed on the bed had dramatic impact upon the flow feature in the jet: a wall jet-like flow was observed before reaching the baffle; after the baffle, a plane jet-like was formed and then the jet flow was deflected toward the water surface along a curvilinear trajectory. Successively, experimental studies for a wall jet impinging onto a forward-facing step in a cross-flow were comprehensively carried out by Langer et al. [18] using planar laser induced fluorescence (PLIF). They presented the jet flow regime after the initial impingement and found that the perimeter and aspect ratio of the jet were dependent of jet-step distance, height of the step and Reynolds numbers. Additionally, predictive correlations for the shape and size of the jet after impingement were discussed. In these two investigations, although the baffle or step was positioned away from the jet exit, their heights were relatively low and submerged in water. Consequently, the jet flow development was not highly confined by the baffle.

The present investigation is to focus mainly on the mean flow development of the 3D confined wall jet. In fact, the 3D confined wall jet plays a significantly important role in hydraulic engineering related to its powerful efficacy for enhancing energy dissipation. The filling and emptying system of navigation lock is a typical case of hydraulic structure producing 3D confined wall jet, which issues from rectangular nozzles (side ports) located in longitudinal culvert. In general, the large amount of water energy is dissipated by the interaction among the jet, bounded ambient fluid in the lock chamber and chamber wall so as to provide better mooring conditions for vessels [19,20]. A schematic sketch of the composite flow in current jet is shown in Figure 1, where the deflected streamlines in the lateral and wall-normal directions are clearly illustrated due to the confinement effect. In particular, after the normal impingement of the jet onto the baffle, corner wall jets in the lateral directions [21,22] and upward wall jet are easily generated in the vicinity of the baffle due to the Coanda effect. The figure also servers to define the coordinate system; $x$, $y$ and $z$ represent the longitudinal, wall-normal and lateral directions, respectively; $U_\mathrm{m}$ denotes the local maximum mean velocity; $y_\mathrm{m}$ is the wall-normal location where $U_\mathrm{m}$ occurs; $y_{\mathrm{m}/2}$ is the distance from the bottom wall to the point in the outer layer where the velocity is half of $U_\mathrm{m}$ (called half-height); $z_{\mathrm{m}/2}$ is the lateral location where the velocity has a half value of $U_\mathrm{m}$ (called half-width). It should be noted that the lateral confinement is neglected.

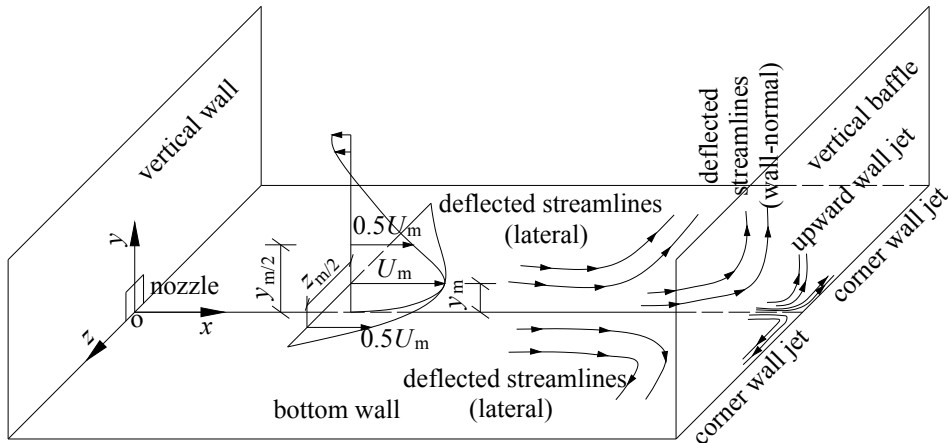

**Figure 1.** Schematic diagram of a three-dimensional wall jet confined by a vertical baffle.

## 2. Experimental Setup

The experiments were performed in a test section which is 800 mm long, 400 mm wide and 400 mm deep. Figure 2 shows schematically the experimental arrangement. The side walls and bottom of the test section were made of clear glass to facilitate the PIV measurements. The wall jet was formed by water passing through a long rectangular pipe, which allowed the flow to fully develop. The pipe has a 14 mm × 16 mm (width × height) cross section and was placed to flush the test section floor. The jet exit velocity was conditioned by a flow control valve and an electromagnetic flowmeter. In this experimental facility, two settling basins attached directly to both ends of the test section were specially designed to condition the flow and obtain the various water depths. Moreover, four constant-head skimming weirs, which were used to stabilize the water surface and ensure the overflow to return back to the supply tank, were constructed and installed at each corner of the settling basins. The Cartesian coordinate system, as shown in Figures 1 and 2, was employed. Note that $x = 0$ is at the jet exit plane, $y = 0$ is on the test section floor, and $z = 0$ is at the symmetry plane of the nozzle.

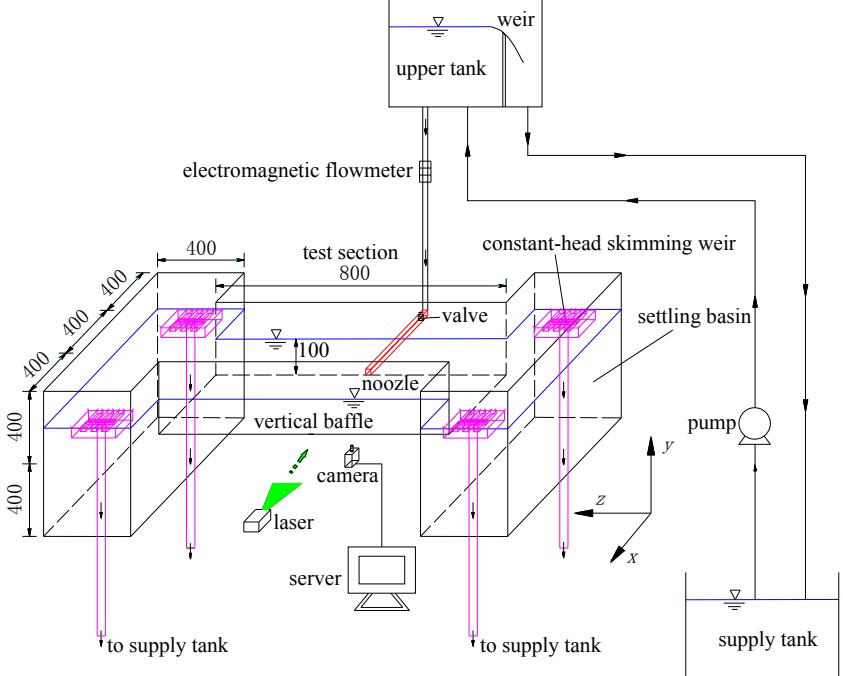

**Figure 2.** A sketch of experimental setup. (All dimensions in mm).

## 3. PIV System and Data Analysis

The velocity field for the jet was acquired using a PIV technique. For the present investigation, the water was seeded with 10 μm hollow glass spheres which have a specific gravity of 1.4. This size and density can ensure that the particles follow the flow synchronously [23,24]. A laser with a continuous energy of 10 W was used to illuminate the flow field. The light sheet formed by the laser was set around 1 mm thick and included the central axis of the jet. All the illuminated images were captured by an 8-bit high-resolution digital camera (NX5-S2 series) with a $2560 \times 1920$ pixels charge-coupled device (CCD). A 50 mm lens (Canon 50 mm f/1.2) was fitted to the camera. All the image pairs were captured at each position with a sampling rate of 1 Hz, a value that is low enough for the images to be uncorrelated [24]. The arrangements of the laser and camera required to be adjusted depending on the plane of measurements. For the *x-z* plane measurements, the laser and camera were positioned at the side and bottom of the test section, respectively. In case of the *x-y* plane measurements, the laser was positioned at the bottom of the test section while the camera was positioned at the side of the settling basin. It should be noted that the location of *x-z* plane measurement depends on the position of $U_{\mathrm{m}}$, and thus varies with the streamwise distance due to the baffle confinement. It is anticipated that the offset *x-z* plane measurements at different *x*-axial locations can describe the vertical variations for mean velocities such as *x*- and *z*-axial velocity, *U* and *W*, respectively. Regarding *x-y* plane measurements, Law and Herlian [11] conducted the offset tests and found that the self-similar velocity profiles still occurred at various sections. Therefore, the offset *x-y* plane measurements from the jet centerline were not performed in the present study.

In the particle image processing, each velocity field involved two consecutive frames, and a time interval between the two frames was critically determined to be 1250 μs such that the maximum particle displacement satisfied the one-quarter rule for PIV correlation analysis [25]. The exposure time for each frame was fixed at 400 μs as a compromise between minimizing image streak and maximizing brightness [26]. The frames presented were divided into numerous small interrogation regions, and the cross-correlation method was used to determine the displacement of the particles in the interrogation window through the peak of the cross-correlation. Subsequently, the local velocity vector of each pair of images was calculated by the displacement and time interval mentioned previously. Detailed information of PIV algorithms is available in the investigation reported by Westerweel et al. [27]. Meanwhile, to improve the computation accuracy for measurements, particle images were processed with the iterative multigrid image deformation method [28]. A three-point Gaussian curve fit was used to determine the peak of displacement with subpixel accuracy. Spurious vectors were removed by the normalized median test method recommended by Westerweel et al. [29] and new vectors were filled by a weighted interpolation approach. The minimum size of the interrogation window of $16 \times 16$ pixels with 50% overlap was used to process the data. The instantaneous image processing program was developed by Beijing Jiang Yi technology co., LTD (Beijing, China). The mean velocity field was calculated using a MATLAB script developed in our laboratory. Considering the effect of the number of instantaneous image pairs on the calculation accuracy of mean velocity, Hu et al. [30] measured the vertical velocity profiles in a high-precision flume, and obtained mean velocities at various vertical locations based on different sample size of image pairs. The mean velocities obtained were compared with that averaged by expected sample size, thereby gaining the standard deviations between the two mean velocities at various vertical locations. In terms of this analysis method, the convergence test for the experimental data, including three mean velocity components at a typical gauge point location, is shown in Figure 3, where 5000 image pairs were selected as the expected sample size. Small enough deviation for *x*-axial mean velocity, *U*, was found after the sample size $N = 2500$, while the corresponding small deviations for *y*- and *z*-axial mean velocities, *V* and *W*, respectively, were obtained at least after $N = 4000$. To ensure faithful mean flow quantities, 5000 PIV instantaneous image pairs were chosen in the present study. In terms of the curve-fitting algorithm for instantaneous vectors, the size of interrogation window, and the required number of instantaneous image pairs for the calculation of the mean velocity, uncertainties in the mean

velocities were estimated to be ±3.4% and ±2.5% for the local velocity close to and away from the wall, respectively.

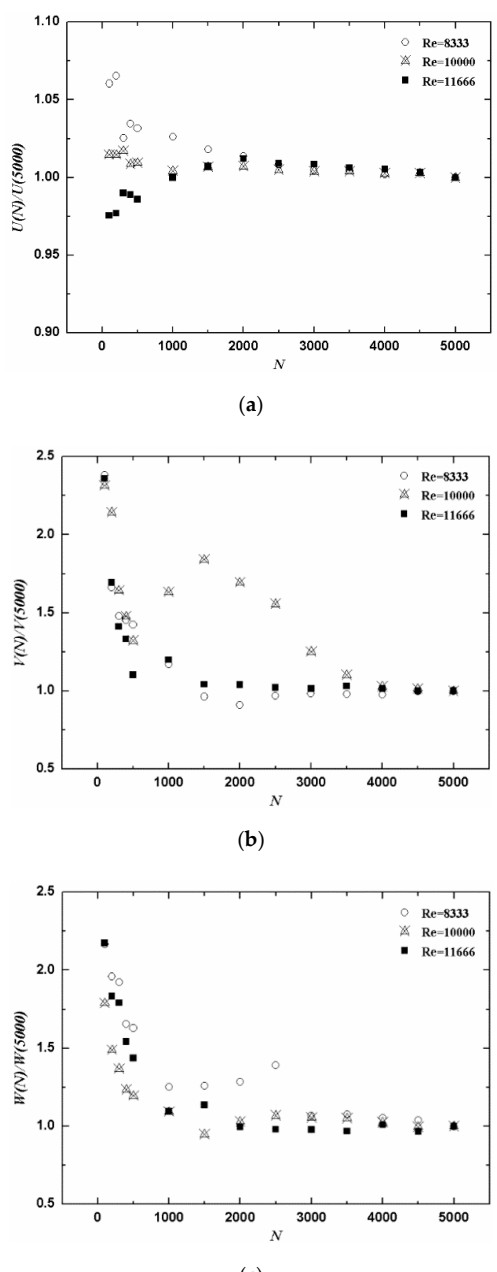

**Figure 3.** Convergence test for the experimental data for three different Reynolds numbers (Re) of 8333, 10,000, and 11,666: (**a**) *x*-axial mean velocity (*U*) varies with the number of instantaneous image pairs (*N*), (**b**) *y*-axial mean velocity (*V*) varies with the number of instantaneous image pairs (*N*), and (**c**) *z*-axial mean velocity (*W*) varies with the number of instantaneous image pairs (*N*).

## 4. Results and Discussion

In the present study, several important mean flow quantities, including velocity profiles and their similarity, growth rate of the half-width and half-height, and decay rate of local maximum velocity, are discussed. Experiments were performed at three Reynolds numbers of 8333, 10,000, and 11,666 (Re = $U_0 B / v$, where $U_0$ is jet exit velocity, $B$ is square root of jet exit area, and $v$ is kinetic viscosity of water) The jet exit velocities of 0.5 m/s, 0.6 m/s and 0.7 m/s, which correspond to the three Reynolds numbers, were determined from the PIV measurements. The corresponding flow rates (*Q*) of 0.089 l/s,

0.109 l/s and 0.131 l/s were measured by the electromagnetic flowmeter. A water depth of 100 mm was set for the test section. All measurements in the lateral ($x$-$z$) and symmetry ($x$-$y$) planes are presented at least for the range of $10 \leq x/B \leq 24$. It should be noted that the measurement locations have not been extended to both the wall and water surface considering the effect of reflection of the laser light on the accuracy of PIV data. The corresponding distances from the measurement edge to the wall and water surface are 1 mm and 5 mm, respectively.

### 4.1. Spreading Rates

Figure 4 shows the variations of the velocity half-height $y_{m/2}$ and half-width $z_{m/2}$ with downstream distance. More specifically, $y_{m/2}$ and $z_{m/2}$ are the wall-normal and lateral locations where $0.5U_m$ occurs, respectively. In this figure, they were normalized by the square root of jet exit area ($B$) which is an appropriate scaling parameter as suggested by Padmanabham and Gowda [10] and Agelin-Chaab and Tachie [13]. The results for unconfined case obtained by Law and Herlian [11] are also included for comparison. The half-height increases approximately linearly in the region $16 < x/B < 23$, but after $x/B = 23$ the value of $y_{m/2}$ grows dramatically (Figure 4a). This behavior may be closely related to a clockwise vortex formed in the region $y/y_{m/2} < 0.25$ as mentioned in Section 4.2. Therefore, the position of $U_m$ tends to be deflected away from the bottom wall. The half-width starts to spread after $x/B = 6$ and varies nearly linearly with downstream distance in the region $6 < x/B < 23$ (Figure 4b). However, the $z_{m/2}$ in the region $16 < x/B < 23$ develops more rapidly compared to the early region $6 < x/B < 16$. Beyond $x/B = 23$, although the confinement of the baffle can enhance the development of the jet flow field, the spreading of $z_{m/2}$ tends to significantly decrease, which is contrary to the variation of $y_{m/2}$ in the corresponding region. This is because the value of $U$ close to the baffle gets considerably dropped due to most of the impinged jet fluid moving in both the lateral directions. In general, the variations of $y_{m/2}$ and $z_{m/2}$ are independent of Reynolds number within the present range.

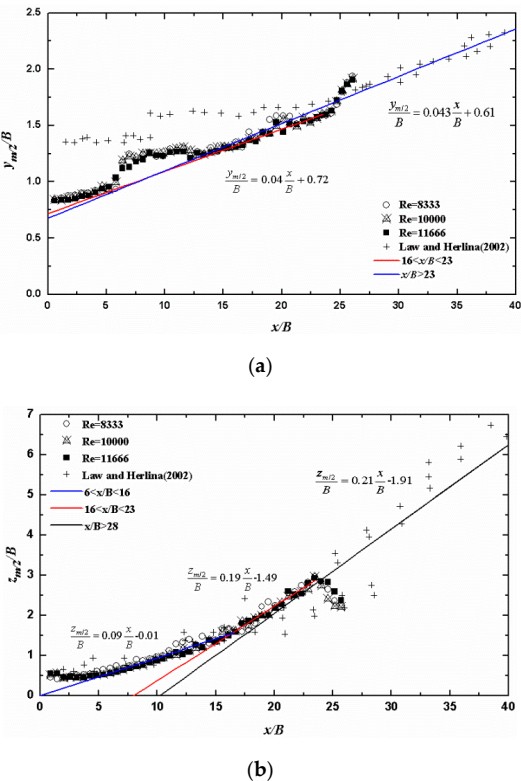

**Figure 4.** Mean flow developments of the 3D confined wall jet with Re = 8333, 10,000 and 11,666: (**a**) velocity half-height $y_{m/2}$, and (**b**) velocity half-width $z_{m/2}$.

To estimate the growth rate ($dy_{m/2}/dx$), a linear fit is applied to the data in the region $16 < x/B < 23$ in Figure 4a. The variation of $y_{m/2}$ can be well described by the equation:

$$\frac{y_{m/2}}{B} = 0.04\frac{x}{B} + 0.72 \tag{1}$$

Similarly, the two linear relationships for $z_{m/2}$ (Figure 4b) in the regions $6 < x/B < 16$ and $16 < x/B < 23$ can be well fitted, respectively. These two equations are written as:

$$\frac{z_{m/2}}{B} = 0.09\frac{x}{B} - 0.01 \tag{2}$$

$$\frac{z_{m/2}}{B} = 0.19\frac{x}{B} - 1.49 \tag{3}$$

Therefore, the corresponding spreading rate $dy_{m/2}/dx$ is 0.04. The $dz_{m/2}/dx$ value of 0.09 in the region $6 < x/B < 16$ is comparable to the circular free jet. Further downstream ($16 < x/B < 23$), the lateral spreading becomes to diverge more rapidly and its slope ($dz_{m/2}/dx$) is 0.19. For comparison, some of previous investigations for 3D circular wall jets are illustrated in Table 1. It is interesting to see that, in the region $16 < x/B < 23$, the spreading rates of $y_{m/2}$ and $z_{m/2}$ for the confined wall jet in the present study, respectively, fall well within the ranges of 0.036 [31]–0.045 [10] and 0.17 [32]–0.33 [31] corresponding to the values in RD region for undisturbed jet reported in the literature. It should be pointed out that reaching RD region for the unconfined wall jet requires streamwise distance at least $x/B = 20$, as summarized in Table 1. These results imply that the baffle starts to alter the jet flow development after $x/B = 16$ and the fully developed region for the confined case appears to occur at least $4B$ earlier than the unconfined case.

**Table 1.** Illustration of previous studies on three-dimensional wall jet.

| Authors | Measuring Technique | Re | RD Region | $dz_{m/2}/dx$ | $dy_{m/2}/dx$ | $n$ |
|---|---|---|---|---|---|---|
| Padmanabham and Gowda [10] | HWA | 95,400 | >20$B$ | 0.216 | 0.045 | 1.15 |
| Law and Herlina [11] | PIV | 5500, 12,200, 13,700 | >23$B$ | 0.21 | 0.042 | 1.07 |
| Agelin-Chaab and Tachie [12,13] | PIV | 5000, 10,000, 20,000 | >60$B$ | 0.255 | 0.054 | 1.15 |
| Després and Hall [16] | PIV | 108,000 | >45$B$ | 0.25 | 0.047 | - |
| Present data | PIV | 8333, 10,000, 11,666 | 16$B$–23$B$ | 0.19 | 0.040 | 1.11 |

Note: $B$ = square root of jet exit area; RD is the radial decay region.

## 4.2. Mean Velocity Profiles

The time-averaged velocity profiles at selected $x/B$ locations in both the vertical and lateral planes are summarized in this section and compared with the previous results, including the free jet [33], two-dimensional [34] and three-dimensional [11,13] wall jets. All the velocities were normalized by the local maximum mean velocity, $U_m$.

Figure 5 shows the streamwise development of the axial ($U$) and vertical ($V$) mean velocity profiles measured in the symmetry plane for Re = 11,666. The $y$ coordinate was normalized by the velocity half-height $y_{m/2}$. The profiles of $U$ collapse reasonably well in the region $10 < x/B < 23$, while the quality of collapse at $x/B = 10$ is relatively poor because the exit flow could not fully develop in the vertical direction, as shown in Figure 5a. Additionally, in the region $y/y_{m/2} < 1.5$, the present data are comparable to the unconfined 2D and 3D wall jet results from Verhoff [34] and Agelin-Chaab and Tachie [13], respectively. Some slight fluctuations of the profiles are observed as the jet evolves downstream to $x/B = 24$ near the baffle. This behavior is attributed to the confinement of the baffle. Further downstream, the confinement becomes more noticeable and negative values of $U$ are observed in the region $y/y_{m/2} < 0.25$. This occurs because most of the impinged jet fluid moves in both the lateral directions and resulting low momentum in the vertical direction could not

overcome the adverse pressure gradient. As a result, a clockwise vortex is formed in the corner. It also can be seen from Figure 5a, near the baffle ($x/B \geq 24$), there are some significant deviations which occur just approximately from $y/y_{m/2} = 1.5$ to 3 due to the resulting reverse flow in the vicinity of the water surface. The deviations could be supported by profiles of wall-normal mean velocity ($V$) in the symmetry plane shown in Figure 5b. After the normal impingement, flow separation occurs close to the baffle and the flow is divided into corner jets [21,22] in both the lateral directions and upward wall jet along the baffle surface due to the Coanda effect, followed by most positive values of $V$ beyond $x/B = 23$ shown in Figure 5b. As expected, the reverse flow is formed after impingement of the upward jet onto the water surface. For the region $21 \leq x/B \leq 24$, some negative values of $V$ are observed in the region $y/y_{m/2} > 3.4$ due to the reverse flow. In the range of $x/B \leq 23$, the magnitudes of $V$ are negative over most of the water depth ($y/y_{m/2} < 3$), indicating that the ambient fluid is being drawn towards the bottom wall, owing to the presence of a secondary mean vortex presented by Launder and Rodi [1]. In addition, compared to the measurements made by Law and Herlian [11] and Agelin-Chaab and Tachie [13], similar variations of $V$ with water depth are observed but the values are slightly lower as illustrated in Figure 5b. However, their measurements reported were selected at least after $x/B = 28$ where the jet flow has been fully developed. It should be noted that considerable scattered points are shown in this figure due to the low accuracy of PIV in the wall-normal direction as described by Law and Herlian [11].

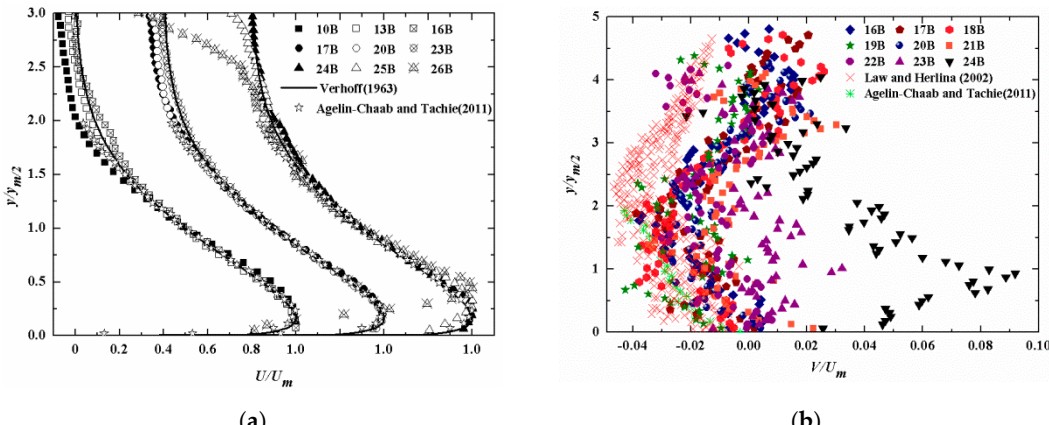

**Figure 5.** Mean velocity profiles measured in *x*-*y* plane for Re = 11,666: (**a**) *U* profiles, and (**b**) *V* profiles.

To further compare the present observations with analytical and previous results, a quantitative evaluation method with mean absolute relative error (MARE [35]) was used in the present study. The MARE is written as:

$$\text{MARE} = \frac{1}{M} \sum_{i=1}^{M} \left| \frac{U - U_p}{U_p} \right| \times 100 \tag{4}$$

where $M$ is total number of data on one velocity profile in given region, $U$ is present *x*-axial mean velocities, $U_p$ is analytical or previous *x*-axial mean velocities. The results of MARE for $U$ profiles in *x*-*y* plane at different locations are summarized in Table 2. In the region ($y/y_{m/2} < 1.5$) unconfined by the water surface, the maximum error is substantially within the range of MARE < 5% before $x/B = 24$, while the maximum error increases to 18.7% after $x/B = 24$. The relatively large error between the confined and unconfined cases indicates that the baffle confinement has noticeable impact upon the mean velocity distribution.

The profiles of axial ($U$) and lateral ($W$) mean velocities at selected locations in the lateral plane are shown in Figure 6, where the $z$ coordinate was normalized by the half-width ($z_{m/2}$). Being similar to the $U$ velocity distribution in the symmetry plane, measurements ($10 < x/B < 23$) in the lateral plane show reasonable collapse in the region $z/z_{m/2} < 1.2$. However, when the flow evolves downstream ($x/B > 16$), there are some slight differences between the experimental data and previous observations

(Figure 6a). This is critically because the profiles of current jet start to be affected by the confinement of the vertical baffle after $x/B = 16$. Far downstream ($x/B \geq 24$), the confinement increases with increasing longitudinal distance and the velocity profiles across the entire sections seem to be unstable. For example, some significant fluctuations can be observed especially after $x/B = 26$ due to the presence of the baffle. For comparison, the results obtained by Law and Herlian [11] generally agree better with the present data. Similarly, Table 3 gives the MARE values for $U$ profiles in $x$-$z$ plane at different locations. Except for the region very close to the baffle ($x/B \geq 26$), the maximum error between the present data and analytical and previous results ($z/z_{m/2} < 1.2$) does not exceed the range of MARE < 5%. Figure 6b shows the $W$ distribution at typical $x/B$ locations. The lateral mean velocity ($W$) increases from zero at the symmetry plane to a peak value which occurs approximately at $z/z_{m/2} = 1.2$ within the region $x/B \leq 24$. Beyond $x/B = 24$, the location of the peak value is gradually delayed (i.e., $z/z_{m/2} = 1.6$ for $x/B = 25$). From this figure, the $W$ profiles in the region $x/B \leq 21$ are relatively lower than those of Law and Herlian [11]. However, as the jet leaves the nozzle, the corresponding $W$ values continuously increase. For example, the present observation at $x/B = 22$ is comparable to those of Law and Herlian [11]. When the jet develops downstream ($x/B = 24$), larger values of $W$ can be observed compared to the results reported by Law and Herlian [11]. Further downstream ($x/B > 24$), the $W$ profiles increase dramatically and are significantly higher as compared to previous results. This indicates that most of the fluid is deflected away from the centerline of the jet in the lateral plane due to the confinement of the vertical baffle.

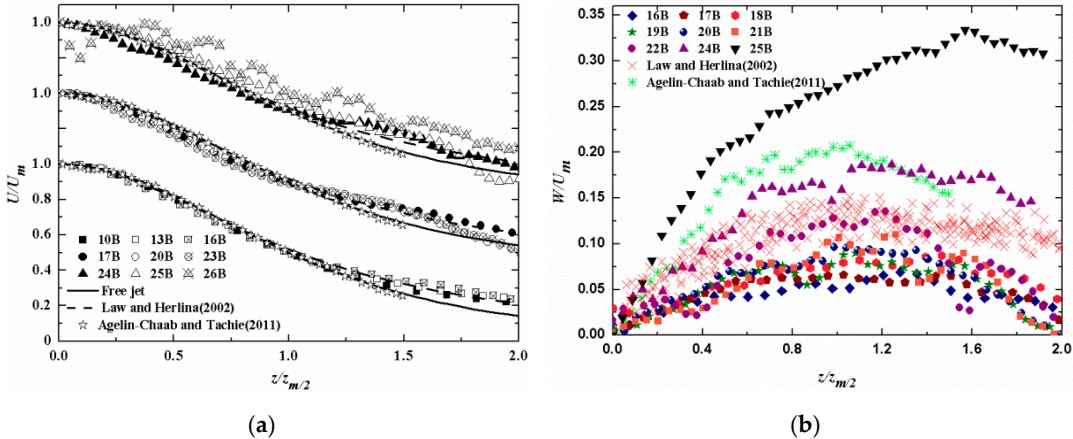

(**a**)　　　　　　　　　　　　　　　　　　　　　　　　(**b**)

**Figure 6.** Mean velocity profiles measured in $x$-$z$ plane for Re = 11,666: (**a**) $U$ profiles, and (**b**) $W$ profiles.

In order to examine the effect of the baffle on the mean velocity profiles more closely, detailed velocity distribution with three Reynolds numbers are plotted in Figure 7. Collapsed curves also can be observed both in the lateral and vertical directions especially within the range of $10 \leq x/B \leq 23$. This indicates that the profiles become independent of Reynolds numbers except the regions near the baffle and water surface.

**Table 2.** MARE (%) values for $U$ profiles in $x$-$y$ plane at different locations ($y/y_{m/2} < 1.5$).

| Location | 10B | 13B | 16B | 17B | 20B | 23B | 24B | 25B | 26B |
|---|---|---|---|---|---|---|---|---|---|
| Analytical solution [34] | 4.13 | 1.45 | 2.63 | 1.30 | 0.92 | 1.40 | 2.75 | 4.65 | 18.70 |
| Agelin-Chaab and Tachie [13] | 4.43 | 2.54 | 3.08 | 1.60 | 2.24 | 2.59 | 3.87 | 5.12 | 18.44 |

**Table 3.** MARE (%) values for $U$ profiles in $x$-$z$ plane at different locations ($z/z_{m/2} < 1.2$).

| Location | 10B | 13B | 16B | 17B | 20B | 23B | 24B | 25B | 26B |
|---|---|---|---|---|---|---|---|---|---|
| Analytical solution [33] | 1.19 | 1.24 | 2.85 | 3.32 | 3.94 | 4.72 | 5.56 | 6.54 | 16.22 |
| Law and Herlian [11] | 2.06 | 1.83 | 2.73 | 3.11 | 3.05 | 3.74 | 5.47 | 5.58 | 15.19 |
| Agelin-Chaab and Tachie [13] | 1.44 | 1.04 | 2.44 | 3.00 | 3.50 | 4.14 | 5.38 | 6.34 | 15.86 |

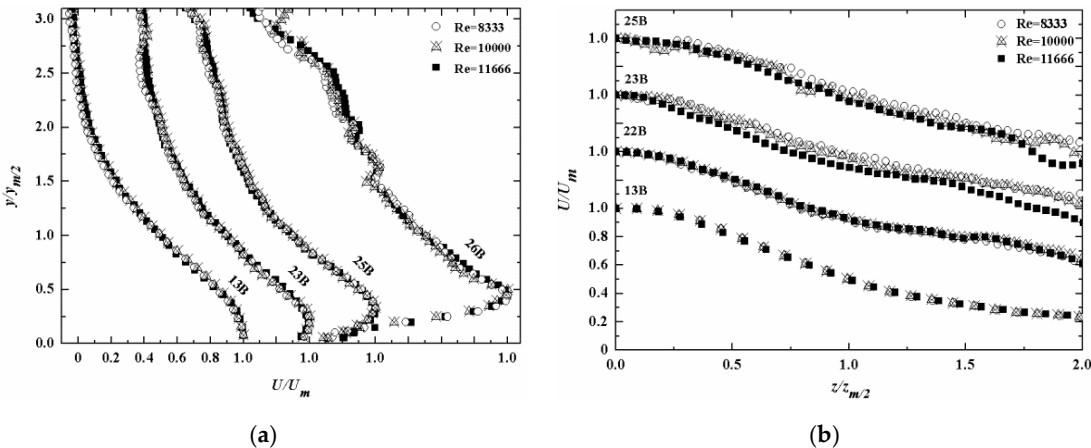

**Figure 7.** Variations of *U* profiles at typical locations with Re = 8333, 10,000 and 11,666: (**a**) *x-y* plane, and (**b**) *x-z* plane.

### 4.3. Decay of Local Maximum Velocity

Further insight into the development of mean velocity can be made by examining how the maximum mean velocity varies with downstream distance, as shown in Figure 8. The value of $U_m$ virtually remains constant in the PC region ($x/B < 3.75$), while it starts to decrease gradually after $x/B = 6$. The $U_m$ decay in both the regions $6 < x/B < 16$ and $16 < x/B < 23$, respectively, can be expressed in power-law forms:

$$\frac{U_m}{U_0} = 2.74\left(\frac{x}{B}\right)^{-0.58} \tag{5}$$

$$\frac{U_m}{U_0} = 11.67\left(\frac{x}{B}\right)^{-1.11} \tag{6}$$

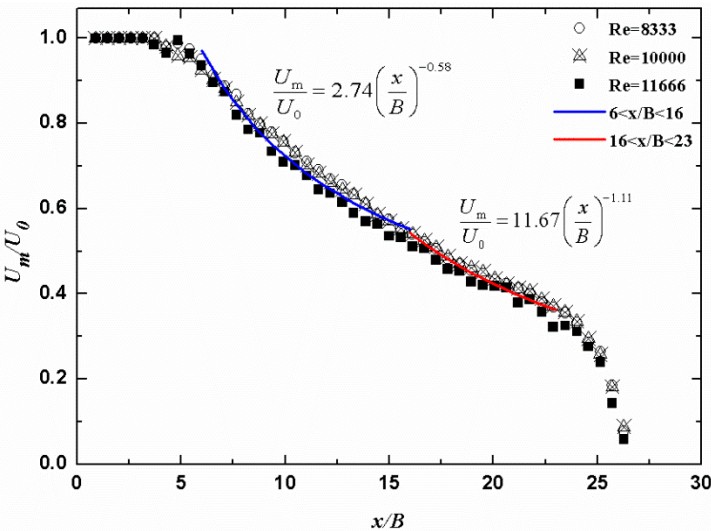

**Figure 8.** Decay of local maximum velocity $U_m$.

The decay rate of 0.58 represents the region $6 < x/B < 16$, which is comparable to the values in the CD region for unconfined 3D wall jets given by Padmanabham and Gowda [10]. In the region $16 < x/B < 23$, significant decay of $U_m$ can be clearly seen from Figure 8 and the corresponding decay exponent is 1.11, a value that is compared very well with those of unconfined cases in the RD region summarized in Table 1. These results are consistent with the previous observations mentioned above.

However, it is anticipated that a sharp reduction of $U_m$ can be observed by the strong confinement of the baffle as the jet develops further downstream ($x/B > 24$).

## 5. Conclusions

An experimental study on a three-dimensional confined wall jet was conducted to examine the effect of a fixed vertical baffle on the mean flow development. Measurements of the flow field were performed using a particle image velocimetry method. The results showed that the confined jet remained unaffected by the baffle until $x/B = 16$, while the other regions of the mean flow development are significantly characterized by the baffle confinement.

In the region ($16 < x/B < 23$), the relatively larger growth rates of 0.04 and 0.19 in the wall-normal and lateral directions, respectively, were obtained for the confined case. These spreading rates fall well within the range of values which correspond to the results in the radial decay region for the undisturbed case. Both the vertical and lateral spreading rates of current jet were found to be independent of Reynolds number. The decay rate was estimated to be 1.11, a value that is consistent with those reported for undisturbed cases in the literature. Similarly, the decay of the maximum mean velocity is independent of Reynolds number. The measurements of mean velocity profiles also exhibit self-similarity and are strongly independent of Reynolds number. Therefore, the fully developed flow of current jet appears to form at least $4B$ earlier than the unconfined case.

As the jet evolves further downstream ($x/B \geq 24$), the confinement becomes more noticeable, and corner jets and upward wall jet are generated close to the baffle. It was found that the jet flow developed more rapidly in the wall-normal direction due to the presence of the baffle. On the contrary, the lateral spreading is significantly reduced. In general, the present mean velocity profiles show reasonable collapse but some differences between the confined and undisturbed jets in the regions very close to the baffle and near the water surface.

It can be concluded that the baffle helps the jet to develop especially beyond $x/B = 16$. This study contributes to a better understanding of the energy dissipation mechanism of current jet. Given the limited measurement cases in the present study, additional tests over considerable distances from nozzle to baffle are needed to fully investigate the mean flow development of the three-dimensional confined wall jet.

**Author Contributions:** M.C. led the work performance and wrote the article; H.H., X.Z., S.L. and R.L. conducted the experiments and collected data through review of papers.

**Funding:** This research was supported by the National Key R&D Program of China, Grant number 2016YFC0402001, National Natural Science Foundation of China, Grant number 51509027, and Key Laboratory of Navigation Structure Construction Technology of China, Grant number Yt918002.

**Conflicts of Interest:** The authors declare no conflicts of interest.

## Notation

| | |
|---|---|
| $B$ | square root of jet exit area |
| $N$ | number of instantaneous image pairs |
| $M$ | total number of data on one velocity profile in given region |
| $n$ | exponent describing the decay of $U_m$ |
| $Q$ | flow rate through the jet pipe |
| Re | jet exit Reynolds number based on jet exit velocity and square root of jet exit area |
| $U$ | $x$-axial mean velocity |
| $V$ | $y$-axial mean velocity |
| $W$ | $z$-axial mean velocity |
| $U_0$ | jet exit velocity |
| $U_m$ | local maximum mean velocity |
| $U_p$ | analytical or previous $x$-axial mean velocity |
| $x$ | longitudinal direction in the coordinate system |
| $y$ | wall-normal direction in the coordinate system |
| $y_m$ | wall-normal location where $U_m$ occurs |

## Notation

| | |
|---|---|
| $y_{m/2}$ | wall-normal location where $0.5U_m$ occurs |
| $z$ | lateral direction in the coordinate system |
| $z_{m/2}$ | lateral location where $0.5U_m$ occurs |
| $\upsilon$ | kinetic viscosity of water |

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
