# Peer review of "Experimental Investigation on Mean Flow Development of a Three-Dimensional Wall Jet Confined by a Vertical Baffle"

_water, doi:10.3390/w11020237_

Round 1
Reviewer 1 Report
Reviewer Blind Comments to Author
The authors tried to measure mean flow development for a 3D rectangular wall jet confined by a vertical baffle. Experiments were performed at three different Reynolds numbers of 8333, 10000 and 11666 based on jet exit velocity and square root of jet exit area. The baffle increases the decay and spreading of the mean flow. Overall, results indicate that the present method can well matched with previous results and can provide information about the energy dissipation mechanism of confined wall jet. However, the following points need to be addressed.
1. As representative parameter, the authors used the mean velocity. However, confined wall jet is 3D and 2D information is difficult to directly get the mean flow. The detailed assumptions and processes should be described in the manuscript.
2. If authors measured 2D PIV for xy and xz planes, authors can represent the 3D flow vectors.
3. It is better to compare the mean velocity with the flowmeter.
4. The authors mentioned the sampling rate of 1 Hz. Information about the frame rate, image processing, and PIV algorithm is restrictive.
5. The authors tried to compare the present results with analytical and previous results. It is better to calculate the coefficient between them.
Author Response
Dear Editors and Reviewers:
Thank you for your letter and for the reviewers’ comments concerning our manuscript entitled “Experimental Investigation on Mean Flow Development of a Three-dimensional Wall jet Confined by a Vertical Baffle” (ID: water-424788). Those comments are all valuable and very helpful for revising and improving our paper, as well as the important guiding significance to our researches. We have studied comments carefully and have made correction which we hope meet with approval. Revised portion are marked in red in the paper. The main corrections in the paper and the responds to the reviewer’s comments are as flowing:
Response to Reviewer 1 Comments
Point 1: As representative parameter, the authors used the mean velocity. However, confined wall jet is 3D and 2D information is difficult to directly get the mean flow. The detailed assumptions and processes should be described in the manuscript.
Response 1: We are very sorry for our negligence of describing the detailed measurement method. It is really true as Reviewer suggested that confined wall jet is 3D and 2D information is difficult to directly get the mean flow. Considering the Reviewer’s comments, we have added the statement of measurement method to revised paper (seeing line 122-128 marked in red): It should be noted that the location of x-z plane measurement depends on the position of Um, and thus varies with the streamwise distance due to the baffle confinement. It is anticipated that the offset x-z plane measurements at different x-axial locations can describe the vertical variations for mean velocities such as x- and z-axial velocity, U and W, respectively. Regarding x-y plane measurements, Law and Herlian [11] conducted the offset tests and found that the self-similar velocity profiles still occurred at various sections. Thus, the offset x-y plane measurements from the jet centerline were not performed in the present study.
In addition, the two consecutive frames were separated by such a short time interval (1250 µs) that reliable mean velocity can be obtained. Detailed information for PIV algorithm can be found in line 129-144 marked in red in revised paper.
As stated above, the mean flow for current jet can be obtained from 2D information.
Point 2: If authors measured 2D PIV for xy and xz planes, authors can represent the 3D flow vectors.
Response 2: To be frank, the measurements by 2D PIV can not represent the 3D flow vectors. But for the x-z plane measurement in the present study, we carried out the corresponding offset tests according to the location of Um. It indicates that the U and W distribution were conducted not only in the lateral plane, but in the vertical plane. However, the offset x-y plane measurements were not performed. This is because the collapsed velocity profiles still work well at various sections. Detailed statement can be also found in line 122-128 marked in red in revised paper.
Point 3: It is better to compare the mean velocity with the flowmeter.
Response 3: Considering the Reviewer’s suggestion, we have added the description of test conditions to revised paper (seeing line 176-180 marked in red): Experiments were performed at three Reynolds numbers of 8333, 10000, and 11666 (Re = U0B/υ, where U0 is jet exit velocity, B is square root of jet exit area, and υ is kinetic viscosity of water) The jet exit velocities of 0.5 m/s, 0.6 m/s and 0.7 m/s, which correspond to the three Reynolds numbers, were determined from the PIV measurements. The corresponding flow rates (Q) of 0.089 l/s, 0.109 l/s and 0.131 l/s were measured by the electromagnetic flowmeter.
In this description, we specified the values of jet exit velocity and corresponding flow rates through the jet pipe, hence the clear relationship between them.
Point 4: The authors mentioned the sampling rate of 1 Hz. Information about the frame rate, image processing, and PIV algorithm is restrictive.
Response 4: We are very sorry for our negligence of providing the information about the frame rate, image processing, and PIV algorithm. Considering the Reviewer’s comments, we have added the corresponding statement to revised paper (seeing line 129-144 marked in red):
In the particle image processing, each velocity field is involved two consecutive frames, and a time interval between the two frames is critically determined to be 1250 µs such that the maximum particle displacement satisfied the one-quarter rule for PIV correlation analysis [25]. The exposure time for each frame was fixed at 400µs as a compromise between minimizing image streak and maximizing brightness [26]. The frames presented were divided into numerous small interrogation regions, and the cross-correlation method was used to determine the displacement of the particles in the interrogation window through the peak of the cross-correlation. Subsequently, the local velocity vector of each pair of images was calculated by the displacement and time interval mentioned previously. Detailed information of PIV algorithms are available in the investigation reported by Westerweel et al [27]. Meanwhile, to improve the computation accuracy for measurements, particle images were processed with the iterative multigrid image deformation method [28]. A three-point Gaussian curve fit was used to determine the peak of displacement with subpixel accuracy. Spurious vectors were removed by the normalized median test method recommended by Westerweel et al [29] and new vectors were filled by a weighted interpolation approach. The minimum size of the interrogation window of 16 x 16 pixels with 50% overlap was used to process the data. The instantaneous image processing program was developed by Beijing Jiang Yi technology co., LTD.
In addition, the relevant references were added to revised paper, e.g., [25]-[29], seeing line 426-435.
Point 5: The authors tried to compare the present results with analytical and previous results. It is better to calculate the coefficient between them.
Response 5: Considering the Reviewer’s suggestion, we have added the comparison of the present results and analytical and previous results by introducing a quantitative evaluation method (seeing line 258-267, 278-281 and 303-304 marked in red in revised paper): To further compare the present observations with analytical and previous results, a quantitative evaluation method with mean absolute relative error (MARE [35]) was used in the present study. In the region (y/ym/2 < 1.5) unconfined by the water surface, the maximum error is substantially within the range of MARE < 5% before x/B = 24, while the maximum error increases to 18.7% after x/B = 24. The relatively large error between the confined and unconfined cases indicates that the baffle confinement has noticeable impact upon the mean velocity distribution.
Similarly, Table 2 gives the MARE values for U profiles in x-z plane at different locations. Except for the region very close to the baffle (x/B ≥ 26), the maximum error between the present data and analytical and previous results (z/zm/2 < 1.2) does not exceed the range of MARE < 5%.
For spreading rates including the velocity half-height ym/2 and velocity half-width zm/2, MARE was not employed since they normally have wide variation range even though data fall in the same region. The focus is on examining whether or not the value of spreading rate is within the expected range. To this end, the statement have been added to revised paper (seeing line 216-219 marked in red): It is interesting to see that, in the region 16 < x/B < 23, the spreading rates of ym/2 and zm/2 for the confined wall jet in the present study, respectively, fall well within the ranges of 0.036 [31] – 0.045 [10] and 0.17 [32] – 0.33 [31] corresponding to the values in RD region for undisturbed jet reported in the literature.
In addition, the relevant references were added to revised paper, e.g., [31], [32] and [35], seeing line 438-441 and 445-446.
Special thanks to you for your good comments.
We tried our best to improve the manuscript and made some changes in the manuscript. These changes will not influence the content and framework of the paper. And here we did not list the changes but marked in red in revised paper.
We appreciate for Editors/Reviewers’ warm work earnestly, and hope that the correction will meet with approval.
Once again, thank you very much for your comments and suggestions.

Reviewer 2 Report
The reviewer wants to thank the authors for their paper about the experimental investigation focusing on the wall jet including a baffle. He/she has some comments/questions, which should be addressed by the authors:
- Abstract/general, Line (L) 18: the square root of the area is the length of the side of an equal quadrat. Would it not be more accurate to use the hydraulic diameter (4*cross-section divided by the wetted perimeter)? Nevertheless, it would be recommendable to give this value another name and not D, which is always linked with a circle. Maybe B?
- Line 39: Very complex geometries are also used in surge tanks as throttles examples can be found in https://doi.org/10.1080/19942060.2018.1443837and DOI: 10.1080/00221686.2018.1454518 (in Press) And also in Spillways DOI 10.1080/00221686.2010.507347 as well as DOI 10.1061/(ASCE)0733-9429(2001)127:8(663) Maybe this connections can be added to the paper.
- L47: the authors use 3d instead of 3D… please check for further cases
- Section starting with L59: This is very important for the understanding of the results and this could be overread by a fast reader, hence it is in the middle of the literature review. Pleas think about this.
- L101: height is 16 mm?
- L127: the description of the analysis is referenced but the paper is in Chinese. So, it is hard to understand the meaning of Figure 3. It would be good to expand this here also if the danger is, that some things have to be repeated, but now in English.
- L148 general: Please include the Q in l/s and also specify the used characteristic linear dimension for the calculation of Re.
- Section 4: Please introduce ym/2 and so on again here, not only in the introduction.
- L157 How close did the measurement go? Please give the exact distance or the used criterium.
- Results general: 4.2 presents the results of ym2 which are depending on the x-distance but previously this value is used to normalize the velocity profile. Would it not be better to first investigate the variable and after that use it to normalize another one?
- Table 1: citations style has to be corrected.
- Please add a Notation section with all the variables.
The reviewer is looking forward to the corrected version and will read the result section ones more in detail. Thank you.
Author Response
Dear Editors and Reviewers:
Thank you for your letter and for the reviewers’ comments concerning our manuscript entitled “Experimental Investigation on Mean Flow Development of a Three-dimensional Wall jet Confined by a Vertical Baffle” (ID: water-424788). Those comments are all valuable and very helpful for revising and improving our paper, as well as the important guiding significance to our researches. We have studied comments carefully and have made correction which we hope meet with approval. Revised portion are marked in red in the paper. The main corrections in the paper and the responds to the reviewer’s comments are as flowing:
Response to Reviewer 2 Comments
Point 1: Abstract/general, Line (L) 18: the square root of the area is the length of the side of an equal quadrat. Would it not be more accurate to use the hydraulic diameter (4*cross-section divided by the wetted perimeter)? Nevertheless, it would be recommendable to give this value another name and not D, which is always linked with a circle. Maybe B?
Response 1: It is really true as Reviewer suggested that the square root of the area is the length of the side of an equal quadrat. Considering the Reviewer’s suggestion, D was corrected and renamed as B to avoid linking with a circle. The revised portion marked in red can be found throughout the revised paper. However, the square root of jet exit area was still used as a scaling parameter. The reason is stated as: In the previous studies for jet, the square root of jet exit area (B) is usually used to normalize the streamwise distance. For example, Padmanabham and Gowda [10] used the square root of jet exit area (B) to scale the streamwise distance instead of the dimension “h”, which is the distance normal to the plate from one edge of a circular orifice in the diametral plane. They found that B is a better scaling parameter than h. Detailed information can be found in Padmanabham and Gowda [10] and Agelin-Chaab and Tachie [13]. Therefore, the square root of jet exit area was still employed as the length scale and named as B instead of D.
Point 2: Line 39: Very complex geometries are also used in surge tanks as throttles examples can be found in https://doi.org/10.1080/19942060.2018.1443837, https://and https://doi.org/ 10.1080/00221686.2018.1454518 (in Press). And also in Spillways DOI https://doi.org/10.1080/00221686.2010.507347 as well as DOI https://doi.org/10.1061/(ASCE)0733-9429(2001)127:8(663). Maybe this connections can be added to the paper.
Response 2: Considering the Reviewer’s suggestion, these 4 connections were added to revised paper, e.g., [5]-[8], seeing line 38 and 385-392 marked in red in revised paper.
Point 3: L47: the authors use 3d instead of 3D… please check for further cases.
Response 3: We are very sorry for our incorrect writing 3D. Corrections according to the Reviewer’s comments were made, seeing line 44, 46 and 51 marked in red in revised paper.
Point 4: Section starting with L59: This is very important for the understanding of the results and this could be overread by a fast reader, hence it is in the middle of the literature review. Pleas think about this.
Response 4: Considering the Reviewer’s suggestion, we have reversed the sequence of the sections starting with line 59 and line 75 in original submission. Moreover, we have re-written and re-organized local contents, seeing line 58-92 marked in red in revised paper.
Point 5: L101: height is 16 mm?
Response 5: We are very sorry for our negligence of specifying the width and height of the jet exit. It is really true as Reviewer pointed out that the jet height is 16 mm. We have added the information of jet cross-section to revised paper, seeing line 100 marked in red.
Point 6: L127: the description of the analysis is referenced but the paper is in Chinese. So, it is hard to understand the meaning of Figure 3. It would be good to expand this here also if the danger is, that some things have to be repeated, but now in English.
Response 6: We are very sorry for our negligence of analysis method proposed by Hu et al [30]. The expansion with regards to this method have been added to revised paper (seeing line 146-156 marked in red): Considering the effect of the number of instantaneous image pairs on the calculation accuracy of mean velocity, Hu et al [30] measured the vertical velocity profiles in a high-precision flume, and obtained mean velocities at various vertical locations based on different sample size of image pairs. The mean velocities obtained were compared with that averaged by expected sample size, thereby gaining the standard deviations between the two mean velocities at various vertical locations. In terms of this analysis method, the convergence test for the experimental data, including three mean velocity components at a typical gauge point location, is shown in Figure 3, where 5000 image pairs is selected as the expected sample size. Small enough deviation for x- axial mean velocity, U, was found after the sample size N = 2500, while the corresponding small deviations for y- and z-axial mean velocities, V and W, respectively, were obtained at least after N = 4000. To ensure faithful mean flow quantities, 5000 PIV instantaneous image pairs were chosen in the present study.
Point 7: L148 general: Please include the Q in l/s and also specify the used characteristic linear dimension for the calculation of Re.
Response 7: Considering the Reviewer’s suggestion, we have added the description of test conditions to revised paper (seeing line 176-180 marked in red): Experiments were performed at three Reynolds numbers of 8333, 10000, and 11666 (Re = U0B/υ, where U0 is jet exit velocity, B is square root of jet exit area, and υ is kinetic viscosity of water) The jet exit velocities of 0.5 m/s, 0.6 m/s and 0.7 m/s, which correspond to the three Reynolds numbers, were determined from the PIV measurements. The corresponding flow rates (Q) of 0.089 l/s, 0.109 l/s and 0.131 l/s were measured by the electromagnetic flowmeter.
In this description, we specified the values of jet exit velocity and corresponding flow rates through the jet pipe, hence the clear relationship between them and characteristic linear dimension for the calculation of Re.
Point 8: Section 4: Please introduce ym/2 and so on again here, not only in the introduction.
Response 8: Considering the Reviewer’s suggestion, we have added the description of ym/2 and zm/2 to revised paper (seeing line 188-189 marked in red): More specifically, ym/2 and zm/2 are the wall-normal and lateral locations where 0.5Um occurs, respectively.
Point 9: L157 How close did the measurement go? Please give the exact distance or the used criterium.
Response 9: Considering the Reviewer’s comments, values of exact distance were added to revised paper (seeing line 184-185 marked in red): The corresponding distances from the measurement edge to the wall and water surface are 1 mm and 5 mm, respectively.
Point 10: Results general: 4.2 presents the results of ym/2 which are depending on the x-distance but previously this value is used to normalize the velocity profile. Would it not be better to first investigate the variable and after that use it to normalize another one?
Response 10: Considering the Reviewer’s suggestion, we have reversed the sequence of the sections 4.1 and 4.2 in original submission. Moreover, we have re-written and re-organized local contents, seeing line 186-304 marked in red in revised paper.
Point 11: Table 1: citations style has to be corrected.
Response 11: We are very sorry for our incorrect writing citations style. Corrections according to the Reviewer’s comments were made, seeing the first column in Table 3 marked in red in revised paper.
Point 12: Please add a Notation section with all the variables.
Response 12: Considering the Reviewer’s comments, a Notation section with all the variables was added to revised paper (seeing line 354-373 marked in red):
Notation
B = square root of jet exit area;
N = number of instantaneous image pairs;
M = total number of data on one velocity profile in given region;
n = exponent describing the decay of Um;
Q = flow rate through the jet pipe;
Re = jet exit Reynolds number based on jet exit velocity and square root of jet exit area;
U = x- axial mean velocity;
V = y- axial mean velocity;
W = z- axial mean velocity;
U0 = jet exit velocity;
Um = local maximum mean velocity;
Up = analytical or previous x-axial mean velocity;
x = longitudinal direction in the coordinate system;
y = wall-normal direction in the coordinate system;
ym = wall-normal location where Um occurs;
ym/2 = all-normal location where 0.5Um occurs;
z = lateral direction in the coordinate system;
zm/2 = lateral location where 0.5Um occurs;
υ = kinetic viscosity of water.
Special thanks to you for your good comments.
We tried our best to improve the manuscript and made some changes in the manuscript. These changes will not influence the content and framework of the paper. And here we did not list the changes but marked in red in revised paper.
We appreciate for Editors/Reviewers’ warm work earnestly, and hope that the correction will meet with approval.
Once again, thank you very much for your comments and suggestions.

Round 2
Reviewer 2 Report
The Reviewer wants to thank the authors for their excellent correction and the answers. By checking the paper again, one point got his\her attention, which should be addressed: The coordinate system in Fig 1 and 2 is not identical. In Fig.2 it is not right handed … sorry, that I didn’t spot this in the first report.
The reviewer is looking forward to the publication. Thank you!
Author Response
Dear Editors and Reviewers:
Thank you for your letter and for the reviewers’ comments concerning our revised manuscript entitled “Experimental Investigation on Mean Flow Development of a Three-dimensional Wall jet Confined by a Vertical Baffle” (ID: water-424788). The comment you pointed out is significantly valuable and very helpful for further revising and improving our paper, as well as the important guiding significance to our researches. We have studied the comment carefully and have made correction which we hope meet with approval. Revised portion are marked in red in the paper. The main corrections in the paper and the responds to the reviewer’s comment is as following:
Response to Reviewer 2 Comments (round 2)
Point 1: The coordinate system in Fig 1 and 2 is not identical. In Fig.2 it is not right handed …
Response 1: It is really true as review suggested that the coordinate system in Fig 1 and 2 is not identical and in Fig.2 it is not right handed. We are very sorry for our incorrect defining z-axial direction for the coordinate system in Fig. 2. The correction according to the Reviewer’s comments was made, seeing line 161 (figure 2) marked in red in revised paper.
In addition, the corresponding dimensions for length were added to revised paper, seeing line 162 marked in red. We are very sorry for our negligence of specifying them.
Special thanks to you for your good comments.
We tried our best to improve the manuscript and made some changes in the revised manuscript. These changes will not influence the content and framework of the paper.
We appreciate for Editors/Reviewers’ warm work earnestly, and hope that the correction will meet with approval.
Once again, thank you very much for your comment and suggestion.
